

# Representation of iron aerosol size distributions is critical in evaluating atmospheric soluble iron input to the ocean

Mingxu Liu[1,2], Hitoshi Matsui[1], Douglas S. Hamilton[3], Sagar D. Rathod[4], Kara D. Lamb[5], Natalie M. Mahowald[6]

[1]Graduate School of Environmental Studies, Nagoya University, Nagoya, Japan

[2]College of Environmental Science and Engineering, Peking University, Beijing, China

[3]Department of Marine, Earth, and Atmospheric Sciences, North Carolina State University, Raleigh, NC, USA

[4]La Follette School of Public Affairs, University of Wisconsin-Madison, Madison, WI, USA

[5]Department of Earth and Environmental Engineering, Columbia University, New York, NY, USA

[6]Department of Earth and Atmospheric Science, Cornell University, Ithaca, NY, USA

*Correspondence to*: H.M. (matsui@nagoya-u.jp)

**Abstract.** Atmospheric aerosol deposition acts as a major source of soluble (bioavailable) iron in open ocean regions where it limits phytoplankton growth and primary production. The aerosol size distribution of emitted iron particles, along with particle growth from mixing with other atmospheric components, is an important modulator of its long-range transport potential. There currently exists a large uncertainty in the particle size distribution of iron aerosol, and the role of aerosol size in shaping global soluble iron deposition is thus unclear. In this study, we couple a sophisticated microphysical, size-resolved aerosol model with an iron-speciated and -processing module to disentangle the impact of iron emission size distributions on soluble iron input to the ocean, with a focus on anthropogenic combustion and metal smelting sources. We first evaluate our model results against a global-scale flight measurement dataset for anthropogenic iron concentration and find that the different representations of iron size distribution at emission, as adopted in previous studies, introduces a variability in modeled iron concentrations over remote oceans of a factor of 10. Shifting the iron aerosol size distribution toward finer particle sizes (<1 μm) enables longer atmospheric lifetime (a doubling), promoting atmospheric processing that enhances the soluble iron deposition to ocean basins by up to 50% on an annual basis. Importantly, the monthly enhancements reach 110% and 80% over the Southern Ocean and North Pacific Ocean, respectively. Compared with emission flux uncertainties, we find that iron emission size distribution plays an equally important role in regulating soluble iron deposition, especially to the remote oceans. Our findings provide implications for understanding the effects of atmospheric nutrients input on marine biogeochemistry, including but not limited to iron, phosphorus, and others.



**1. Introduction**
Iron is a critical micronutrient supporting marine primary production, which is closely associated with marine carbon-nitrogen
cycles in the Earth system [*Mahowald et al.*, 2018; *Moore et al.*, 2001]. The atmospheric deposition of soluble iron to many
ocean basins has long been regarded as an important source of bioavailable iron for ocean biota uptake in iron-limited areas
[*Jickells and Moore*, 2015; *Jickells et al.*, 2005; *Tagliabue et al.*, 2017]. Understanding the amount and past-to-future evolution
of atmospheric iron deposition to the ocean is critical in assessing the ocean carbon sequestration under a changing climate
[*Bergas‑Massó et al.*, 2023; *Hamilton et al.*, 2020a; *Myriokefalitakis et al.*, 2020].
The quantification of soluble (bioavailable) iron input to the ocean is linked to differences in iron emission source properties,
the degree to which iron aerosol undergoes acidic or organic chemistry, and atmospheric transport [*Hamilton et al.*, 2022].
Atmospheric iron comes from three major emission sources, i.e., wind-blowing dust, wildfire and biomass burning, and
anthropogenic activities, such as fossil fuel combustion and iron smelting. Dust storms, which frequently occur in arid or semi-
arid regions of the world, such as North Africa and East Asia, provide an abundant iron source to the ocean and support primary
production [*Mahowald et al.*, 2009; *Westberry et al.*, 2023]. In addition, a growing body of evidence is showing that pyrogenic
iron, with higher fractional solubility than dust [*Ito et al.*, 2019], is a large source of atmospheric soluble iron deposition to
many ocean basins, including the Southern Ocean, Northern Pacific Ocean, and Northern Atlantic Ocean [*Conway et al.*, 2019;
*Liu et al.*, 2022; *Matsui et al.*, 2018; *Seo and Kim*, 2023]. Because the strength of each source could be affected by future
climate change and/or human activities, their contributions to bioavailable iron input to the ocean may vary regionally and
temporally by the end of the century [*Bergas‑Massó et al.*, 2023].
Atmospheric transport provides the essential pathway in which iron aerosol emitted from the land is supplied to the remote
ocean. Atmospheric circulation patterns dictate the main transport pathways for aerosol to follow and thereby which source
regions are important to consider in terms of their supply to ocean basins. Additionally, atmospheric transport enables internal
mixing of iron-bearing aerosols with other aerosol and gas components, like sulfates and organics; a process commonly known
as aging that facilitates the dissolution of iron from an insoluble state to a soluble state [*Li et al.*, 2017; *Shi et al.*, 2020; *Shi et
al.*, 2012; *Solmon et al.*, 2009]. The atmospheric aging processes can significantly increase iron solubility and subsequent
soluble iron deposition, evidenced by both in-field and laboratory observations and global model simulations [*Ito*, 2015; *Li et
al.*, 2017; *Longo et al.*, 2016]. Uncovering the underlying mechanisms of the aging processes and associated enhancement of
iron solubility during transport is an ongoing topic of investigation [*Meskhidze et al.*, 2019; *Shi et al.*, 2020].
To elucidate atmospheric flux of iron-containing aerosols to the ocean, atmospheric models have been developed to include a
range of iron emission sources that currently show a large intermodal difference in flux estimates [*Myriokefalitakis et al.*,
2018]. Part of the problem is that it is difficult to realistically reproduce the distribution of soluble iron concentrations across
all the different regions of the world, and especially over the remote polar oceans that are often characterized by low iron
concentrations with a high fractional solubility [*Ito et al.*, 2019]. Note that the size distribution of iron, which is an important
consideration when determining the particle lifetime, and thus its long-range transport potential, is key in atmospheric iron
flux to the ocean [*Hamilton et al.*, 2020b; *Myriokefalitakis et al.*, 2018]. Compared to coarse-sized particles (e.g., larger than
1 μm), smaller particles generally feature lower loss rates with respect to dry deposition and wet removal, resulting in longer
atmospheric lifetimes; being transported a longer distance increases the potential for atmospheric processing (i.e., longer period
of aerosol ageing) and thus higher soluble iron deposition. Representation of iron size distribution in models could be therefore
important.
Iron aerosol characteristics depend in part on differences between source types. The iron mass size distribution associated with
natural dust sources commonly pertains to mineral dust aerosols, with the coarse-sized (diameter greater than 1 μm) fraction



dominant [*Albani et al.*, 2014; *Mahowald et al.*, 2014]. Similarly, fire iron emissions are dominated (>80%) by coarse mode
particles [*Hamilton et al.*, 2019], suggested to be due to the entrainment of local dust iron-bearing aerosol in the pyro convective
updrafts generated by a fire [*Hamilton et al.*, 2022]. For iron aerosol with an anthropogenic source, however, the relative
fractions between the fine and coarse particle size distribution at emission are more divergent among previous investigations.
Recent observational constraints reveal large mass concentrations of anthropogenic iron oxide in the fine mode [*Moteki et al.*,
2017], while a subset of modelling studies have treated most of this iron in the coarse mode [*Wang et al.*, 2015]. As opposed
to those natural sources, anthropogenic iron size distributions may be highly variable with respect to diverse fuels, combustion
temperatures, and industrial processes, as well as the abatement technologies applied to control air pollution [*Hamilton et al.*,
2020b; *Rathod et al.*, 2020].
The extent to which iron aerosol size distributions shape the pattern of atmospheric soluble iron inputs to different ocean
regions is currently unknown. Herein, by leveraging a size-resolved global aerosol model configured with iron processes, we
focus on the representation of anthropogenic iron size distributions at emission, primarily involving its roles in altering global
long-range transport and deposition fluxes of iron. We further put the effect of iron size distribution in the context with iron
emission uncertainty to shed light on their relative importance in controlling global-scale iron deposition.

**2. Methods and Materials**
**2.1 Global aerosol model**
We conducted global aerosol simulations using the Community Atmospheric Model version 5 (CAM5.3) with the Aerosol
Two-dimensional bin module for foRmation and Aging Simulation version 2 (CAM-ATRAS) [*Matsui*, 2017; *Matsui and*
*Mahowald*, 2017]. The model treats a series of aerosol chemical and microphysical processes in a size-resolved manner with
12 aerosol size bins from 1 to 10,000 nm in diameter. Our recent study suggests that this size-revolved method can well
represent the growth of small particles to larger ones and the evolution of particle size distributions during atmospheric
transport [*Liu and Matsui*, 2022]. The improvement of aerosol in-cloud wet scavenging process was included to improve the
modelling of aerosol long-range transport efficiency [*Liu and Matsui*, 2021]. The CAM-ATRAS model has been adequately
validated for aerosol mass and number concentrations at a global scale using comprehensive measurements from the ground
to the upper troposphere [*Gliß et al.*, 2021; *Kawai et al.*, 2021; *Matsui and Liu*, 2021; *Matsui et al.*, 2022; *Matsui and Moteki*,
2020].
To represent the iron cycle from emission to deposition, we explicitly treated iron constituents within the aerosol model, similar
to our previous study [*Liu et al.*, 2022], while with updates for iron processing in the current work. All iron-bearing components
were assumed to be internally mixed with other aerosols and underwent emission, physical transport, chemical aging (e.g.,
solubilization), and deposition in the atmosphere. Our model simulated iron solubility and atmospheric processing of iron-
bearing aerosols through the online coupling with the Mechanism of Intermediate complexity for Modelling Iron [*Hamilton et*
*al.*, 2019]. For anthropogenic iron, we consider five different minerals, namely magnetite, hematite, illite, kaolinite, and sulfate
iron, following the global emission inventory by *Rathod et al.* [2020]. We account for a wide range of anthropogenic sources
including iron smelting and fossil fuel combustion sources taken into account. Dust iron emission was calculated by assuming
a constant iron content of 3.5% in dust aerosol emission [*Shi et al.*, 2012]. The global interannual mean iron (insoluble +
soluble) emissions from dust, biomass burning, and anthropogenic sources were 87 Tg Fe yr$^{-1}$, 1.1 Tg Fe yr$^{-1}$, and 2.2 Tg Fe
yr$^{-1}$, respectively.



We validated our modeled anthropogenic iron oxide concentrations against a global-scale aircraft measurement in the
troposphere consisting of eight campaigns for the periods of 2009−2011 and 2016−2018 [*Lamb et al.*, 2021]. These
observations include iron oxide aerosols with volume equivalent diameters between 180-1290 nm. The model results were
extracted along the flight tracks in time and space and further averaged in several latitudinal bands across Pacific and Atlantic
Oceans (see Fig. S1). More details can be seen in *Liu et al.* [2022].
The model was compiled with a horizontal resolution of 1.9º × 2.5º (latitude × longitude) and 30 vertical layers from the
surface to 40 km. We ran the model for the two periods, 2008−2011 and 2015−2018, with the first year in each period as spin-
up. The meteorological fields were nudged by the Modern-Era Retrospective analysis for Research and Applications Version
2. In addition, to provide implications for ocean biogeochemistry, we estimated the changes in marine net primary production
induced by iron inputs following the methods used by *Rathod et al.* [2022] and *Okin et al.* [2011], in which a cut-off (4 μmol
$L^{-1}$) of nitrate concentrations at surface water was chosen to define the geographical areas of iron-limited ocean basins.

### 2.2 Representation of iron size distribution

The number and mass size distribution of aerosol at emission is an essential parameter in aerosol modelling. However, due to
the limited knowledge about iron emission characteristics, the representation of iron size distributions varies greatly for
anthropogenic sources. We tested four different size distributions of anthropogenic iron at emission (Fig. 1), briefly described
as follows.
First, we adopted the same size distribution with our previous studies [*Liu et al.*, 2022; *Matsui et al.*, 2018] based on *Moteki*
*et al.* [2017], abbreviated as Moteki2017 hereafter, which was derived from a power law function to fit the observed
anthropogenic iron oxides concentrations within the boundary layer in the outflow of East Asian sources. Note that the
observed size-revolved number concentrations were confined within 170 to 2,100 nm in diameter given the detection limit. By
extrapolating the observation results, we obtained the mass size distribution between 1 nm and 10,000 nm with negative values
excluded. In this case, more than 90% of iron mass was allocated to the size range of 100 nm to 2,500 nm. In addition, we
varied the emissions by a factor of two (×0.5 and ×2.0, respectively) in another two parallel experiments to account for the
potential uncertainties in iron emission estimates.
The second case was derived from *Rathod et al.* [2020] (abbreviated as Rathod2020 hereafter). They developed a new
mineralogy‐based iron emission inventory by introducing more details in anthropogenic sources, especially the inclusion of a
metal smelting source. These improvements increase the fine aerosol (less than 1 μm in diameter) fractions by a factor of 10
higher than most previous inventories. Consequently, this inventory was characterized with almost equal fractions between
fine- and coarse-sized emissions, while previous inventories always applied a much larger fraction for the coarse mode.
Consistent with Rathod et al., we allocated 10% and 90% of fine iron mass to the Aitken mode and the accumulation modes
of aerosols, respectively.
The third case was derived from *Hamilton et al.* [2019] (abbreviated as Hamilton2019 hereafter). They revised the
anthropogenic iron emission inventory based on *Luo et al.* [2008] (no metal smelting) and showed that the ratio of fine-sized
iron mass with that of the coarse-sized was 1:5.6, which resulted in the coarse mode dominating. Asimilar ratio for
anthropogenic iron emissions was applied by *Ito* [2013]. Also, 10% of fine-sized emissions were allocated to the Aitken mode
and the remaining 90% to the accumulation mode.
The fourth case was derived from *Wang et al.* [2015] (abbreviated as Wang2015 hereafter). In their combustion-iron inventory,
the ratio of fine-mode mass to the coarse mode was as low as 1:24, because only 0.1–0.3 % of iron mass from coal fly ash
were emitted in the fine mode. Thus, we allocate 96% of iron in the coarse mode and the remaining 4% in the fine mode.




145 To enable the intercomparison among these cases, we used the global-scale anthropogenic iron emission mass inventory from
146 *Rathod et al.* [2020] but with different allocations between fine and coarse sizes in each case. Therefore, the simulated
147 variability in atmospheric iron input to the ocean between cases should be attributable to iron size distributions rather than
148 emission amount. For the last three cases, we adopted three constant log-normal modes to distribute iron emissions, namely
149 Aitken, accumulation, and coarse modes, with their determining parameters including geometric standard deviations, number
150 mode diameter, and density reported by *Hamilton et al.* [2019]. We then separated these three modes into 12 size bins from 1
151 nm to 10,000 nm adapted for our size-resolved aerosol modelling. The size distributions of iron from biomass burning and
152 dust sources were consistent with *Liu et al.* [2022] in all cases. In the following analysis, we grouped the Moteki2017 and
153 Rathod2020 as the fine-sized group while the Hamilton2019 and Wang2015 as the coarse-sized group.

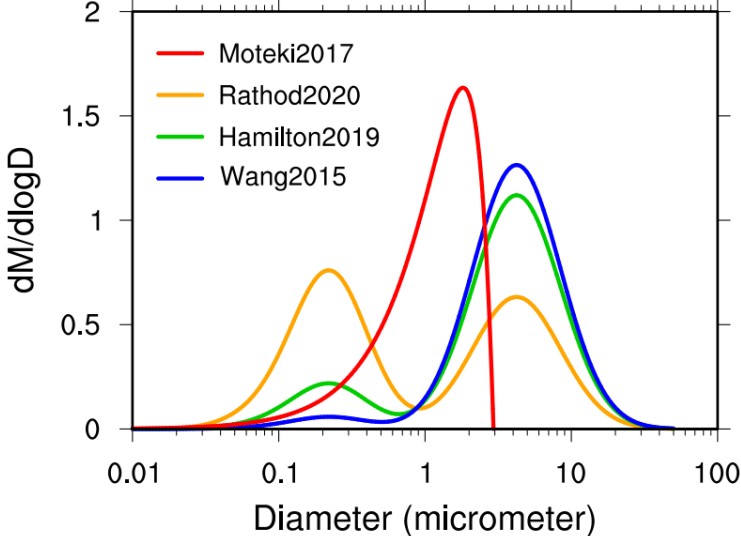

154

**Fig. 1**. Mass size distribution functions for anthropogenic iron emission adopted in four previous studies. The Moteki2017
curve was provided by fitting in-situ measurements for iron particles within the range of 170 and 2,100 nm in diameter over
East Asia; the Rathod2020 was based on the latest combustion iron emission inventory, with updates on iron estimate in fine-
mode sizes (<1μm); the Hamilton2019 and Wang2015 cases were modeling studies that assumed most anthropogenic
combustion in the coarse-mode bin.

160

**3. Results and Discussion**

**3.1 Atmospheric iron aerosol concentrations**

163 We first examined the effect of changes to iron particle size distributions for anthropogenic sources (unless otherwise stated)
164 on the atmospheric iron aerosol burden and its global distributions. Figure 2 illustrates that the global-mean anthropogenic
165 iron lifetimes differ by about a factor of 2 among the four examined cases. Both the Moteki2017 and Rathod2020 cases
166 simulated a lifetime around 3.0 days. In contrast, the Hamilon2019 and Wang2015 cases simulated a lifetime around half as
167 long (between 1.4 and 1.7 days). As only the size distribution is changed in these simulations the change in lifetime is directly
168 linked to the apportionment of mass aerosol between fine and coarse particle size modes. This result is in line with previous
169 reports [*Hamilton et al.*, 2020b], and demonstrates that shifting emitted iron toward fine-sized diminishes atmospheric loss
170 rates of iron aerosols via dry (sedimentation) and wet (precipitation) removal pathways and extends their lifetime.
171 Consequently, given the same emission amount, the atmospheric iron burdens are enhanced, accordingly by approximately a
172 factor of 2 from ~9.0 Gg in the coarse-sized cases (Hamilton2019 and Wang2015) to ~18.0 Gg in the fine-sized cases



(Moteki2017 and Rathod2020). In a similar manner, the lifetimes and mass burdens of anthropogenic iron in the soluble form
are almost doubled in the fine-sized cases (Fig. 2b). The extended lifetimes also enhance the globally averaged iron solubility
(Fig. S2), by allowing more iron subject to aerosol aging and solubilization processes.

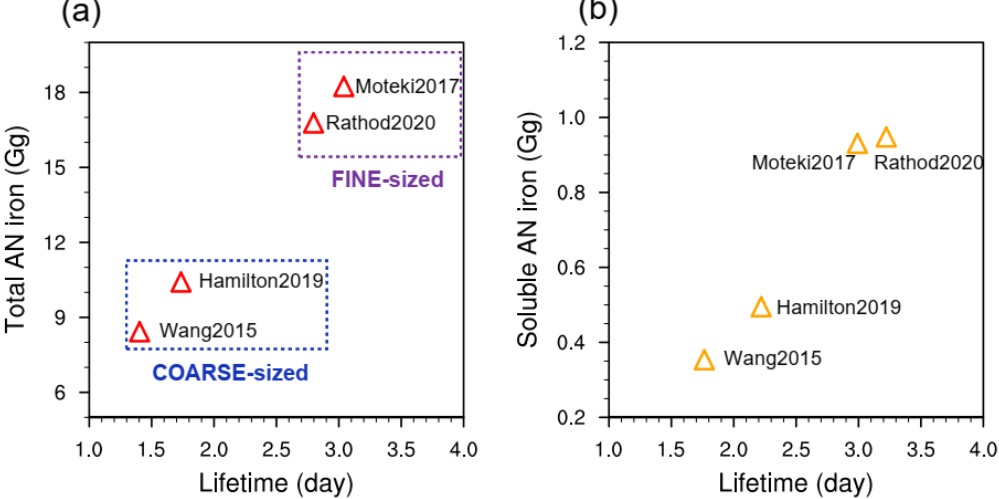


**Fig. 2**. Global anthropogenic iron concentration burdens and lifetimes varied by the emission size distributions. The scatter
plots are shown for (a) anthropogenic iron (labeled as AN iron) burden vs. lifetime and (b) soluble anthropogenic iron vs.
lifetime. Four representative cases are examined in this work: Moteki2017, Rathod2020, Hamilton2019, and Wang2015. The
first two cases are grouped into "FINE-sized" and the other two are grouped into "COARSE-sized".



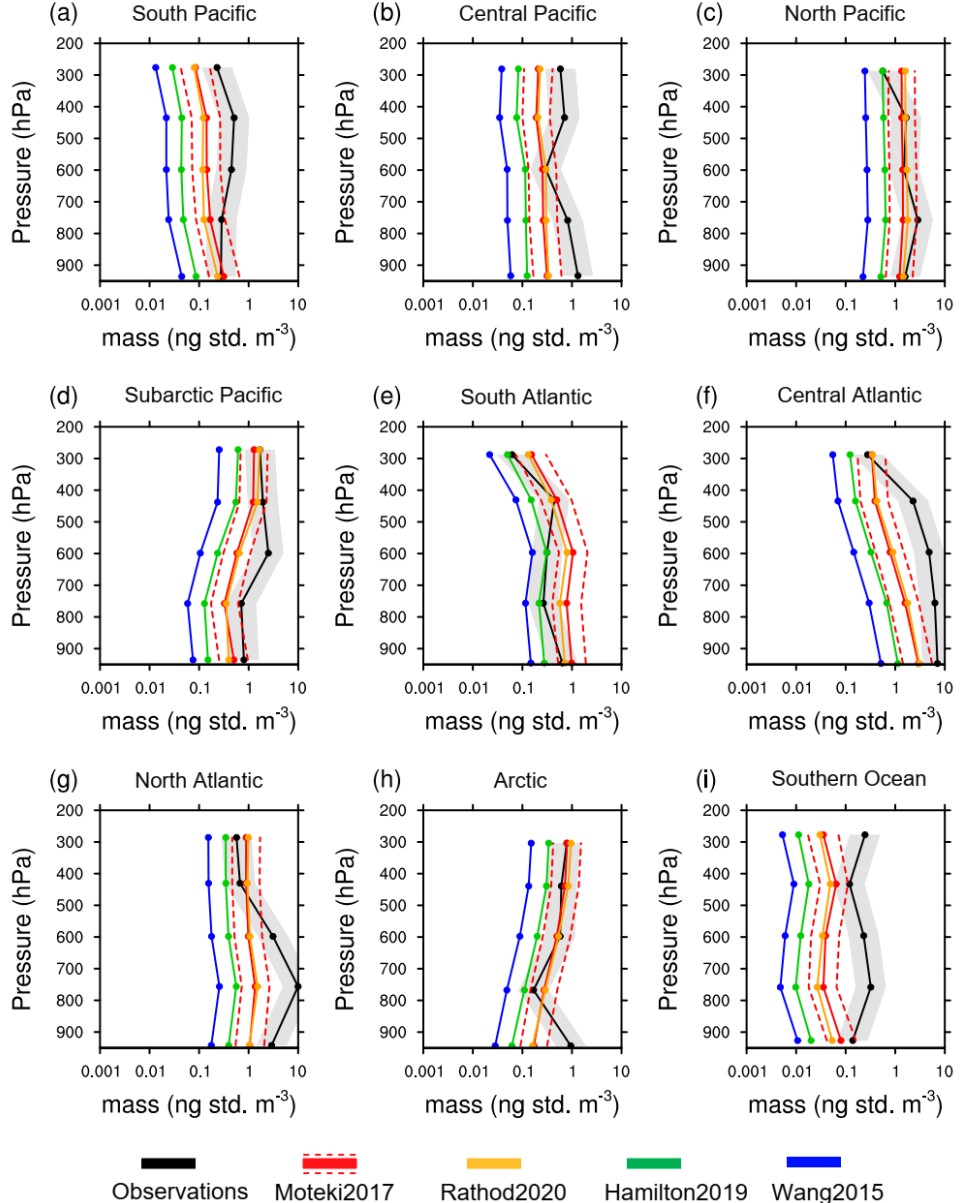


**Fig. 3**. Comparison of modelled anthropogenic iron mineral (magnetite) vertical concentration profiles in four cases with aircraft measurements across global oceans. The flight routes and model-observation sampling methods have been documented in *Lamb et al.* [2021] and *Liu et al.* [2022]. The geographical location of each oceanic area is marked in Fig. S1. We also scaled up and down emission fluxes by a factor of 2 from the Moteki2017 case, respectively, to account for potential uncertainties in emission estimates (red dashed lines in the panels).

To evaluate each iron simulation, we compare simulated aerosol characteristics against global-scale aircraft measurements of
anthropogenic magnetite within nine regions of the troposphere (Fig. 3). The size distribution consistent with the measurements
is selected for our model results. Note that magnetite, comprised about 70% of anthropogenic iron emissions, can be used an
indicator of anthropogenic iron abundance in the atmosphere [*Matsui et al.*, 2018; *Rathod et al.*, 2020].  Despite the same
emission fluxes considered in all cases, their simulated magnetite aerosol concentrations can differ by up to a factor of 10.
Specifically, the Moteki2017 and Rathod2020 cases show a much higher performance in reproducing the observed profiles





over all ocean basins compared to Hamilon2019 and Wang2015 cases. In particular, the cases with a more uniformly
distributed particle size distribution across modes captures the high concentration (>1 ng m$^{-3}$) in North Pacific, which can be
linked to atmospheric plumes transported from East Asia with intensive emission rates [*Moteki et al.*, 2017; *Seo and Kim*,
2023]. Some underestimations still exist in near-surface or high altitudes. Doubling the emission fluxes based the Moteki2017
case can appreciably narrow the gaps with the observation (dashed red lines in Fig. 3). In contrast, the coarse-sized dominated
simulations (i.e., Hamilton2019 and Wang2015) underrepresents the magnetite concentrations over global remote oceans,
particularly by up to one order of magnitude over the Pacific Ocean The shorter lifetimes in this group limit the long-range
transport of iron aerosols from continental sources to the remote atmosphere.
These results imply that agreement between observations and model simulations can be improved by reducing uncertainties in
the emission inventory and in the long-range transport efficiency associated with representation of iron size distributions.
Moreover, as illustrated in Fig. 3, the variabilities (the ratio between the maximum and minimum) of simulated magnetite
vertical profiles by iron size distribution changes are wider than those by the emission uncertainties, for which emission fluxes
were perturbed by a factor of 2 (×2.0 and ×0.5, respectively) to test the sensitivity of simulated iron concentrations. We
therefore highlight that in order to observationally constrain iron emissions more realistically in global aerosol simulations, it
is a prerequisite to use a realistic empirical representation of anthropogenic iron aerosol size distributions.

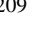

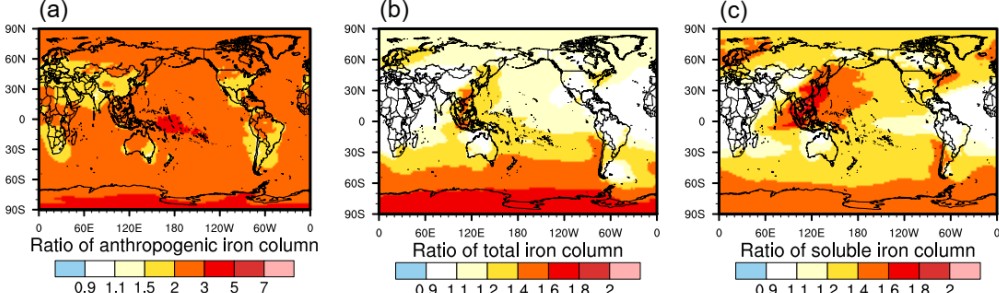


**Fig. 4**. Global map of variability in iron column concentrations between the fine-sized group and the coarse-sized group. The
variability ratios (fine-sized/coarse-sized) are calculated for (a) anthropogenic iron, (b) total iron, and (c) soluble iron,
respectively. Herein, total iron is a combination of iron from dust, biomass burning, and anthropogenic sources, the soluble
form of which denotes soluble iron. The ratios represent the maximum differences between the fine-sized group (the
Moteki2017 and Rathod2020 cases) and the coarse-sized group (the Hamilton2019 and Wang2015 cases) and indicate the
spread of iron simulations.

Our simulations further demonstrate that the representation of iron size distribution shapes the iron aerosol concentrations at a
global scale (Fig. 4). Anthropogenic iron in the fine-sized group shows higher column concentrations by more than a factor of
2 than in the coarse-sized over oceans. The differences (ratios) are larger in those remote oceans compared to source regions
like East Asia, southern Africa and South America, because iron-bearing aerosols in smaller sizes can be transported to a
longer distance. The variability between the two groups is much less relative to total iron (Fig. 4b), which also includes
contributions from dust and biomass burning sources. One exception is the Southern Ocean with still large ratios around 1.5,
reflecting an important fractional contribution of anthropogenic iron among all sources over this region. Note that the
differences for soluble iron are more pronounced than for total iron (Fig 4c), because of the higher solubility of anthropogenic
iron than dust iron. Over East Asia and its outflow areas, the rapid aging process in the polluted environments are capable of
enhancing iron solubility and thus amplify the differences of soluble iron concentrations between the fine-sized and coarse-
sized groups [*Li et al.*, 2017; *Zhu et al.*, 2022].



**3.2 Atmospheric soluble iron input to the ocean**

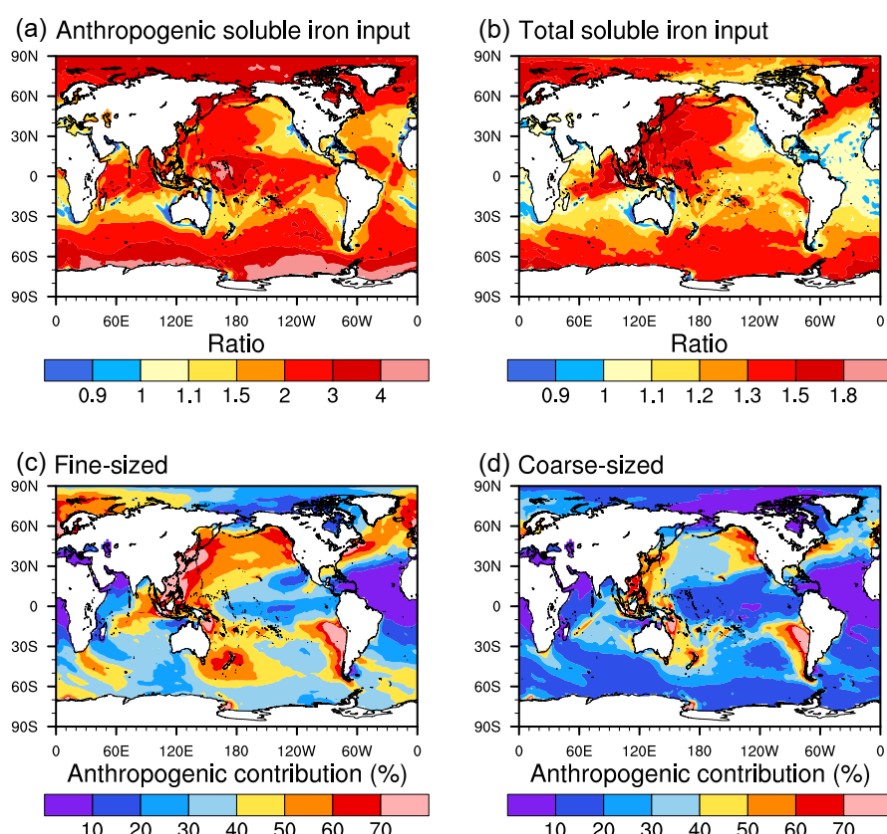


**Fig. 5**. Comparison of atmospheric soluble iron input to the ocean between the fine-sized and coarse-sized cases. Shown here
are (a) the ratio of anthropogenic soluble iron simulated in the fine-sized case to the result of the coarse-sized, (b) the ratio of
total soluble iron, (c) the fractional contribution (in percentage) of anthropogenic emission to total soluble iron input in the
fine-sized case, and (d) the fractional contribution of anthropogenic emission in the coarse-sized case.

Next, we examine the extent to which the iron size distributions at emission can alter soluble iron input to the global ocean
basins, which is vital to net primary productivity, especially in the high-nutrient, low-chlorophyll (HNLC) areas [*Hamilton et
al.*, 2022; *Moore et al.*, 2013]. As illustrated in Fig. 5, though the global emission and the resulting annual iron (insoluble +
soluble) deposition amount are the same between cases, their geographical distributions vary substantially. Using the ratio of
annual soluble iron deposition in the fine-sized group to that of the coarse-sized as a proxy of the variability, we find the fine-
sized distributions lead to much more soluble iron input to remote ocean basins including North Pacific and Southern Ocean,
by up to a factor of 4 for the anthropogenic sources and 1.5 for the total of all sources (i.e., anthropogenic + fire + dust). Similar
spatial patterns emerge regarding the total iron deposition, suggesting the importance of long-range transport efficiency in
regulating iron distributions (Fig. S3). The variability is negligible in the equatorial and subtropical Atlantic, where dust iron
dominates soluble iron input to this area. The ratios less than 1, indicating reduced deposition fluxes in the fine-sized group,
are found near the continental sources, including western U.S., Australia, and southern Africa, because of the slower deposition
speed. However, in East Asia, which has most intensive anthropogenic iron emissions, the shift toward finer sizes also increase
soluble iron deposition near the sources (e.g., eastern China). This is attributable to the rapid aging processes of the fine-sized
iron in such polluted environment that convert more insoluble iron to its soluble form [*Baldo et al.*, 2022; *Zhang et al.*, 2022].



The source appointment of soluble iron deposition across ocean basins also varies with iron size distributions (Fig. 5c-d). The
anthropogenic iron emission becomes more dominant in North Pacific, North Atlantic, and parts of the Southern Ocean, with
its fractional contribution reaching more than 50% in the fine-sized group compared to that of 30–40% in the coarse-sized.
Such variability is attributable to the enhancement of anthropogenic soluble iron fluxes to those remote oceans by the shift of
anthropogenic iron emission toward finer size bins. Globally, the soluble iron deposition from anthropogenic sources is 55.0
Gg per year for the fine-sized group, larger than that of 35.3 Gg per year for the coarse-sized group. Hence, even though the
same emission is applied in these simulations, the diversity of iron size distributions at emission yields a considerable
variability of soluble iron deposition on a global basis. As discussed earlier, the extended iron lifetime by about a factor to 2
in the fine-sized group allows more iron to be transported to a remote region and simultaneously increases the amount of
atmospheric iron processing and dissolution to a soluble form. Of the total iron deposition, the soluble fraction is thus notably
elevated. We also find that the chemical aging process, as the major source of soluble iron, controls the differences of soluble
iron deposition over remote oceans between the fine-sized and coarse-sized groups (Fig. S4).

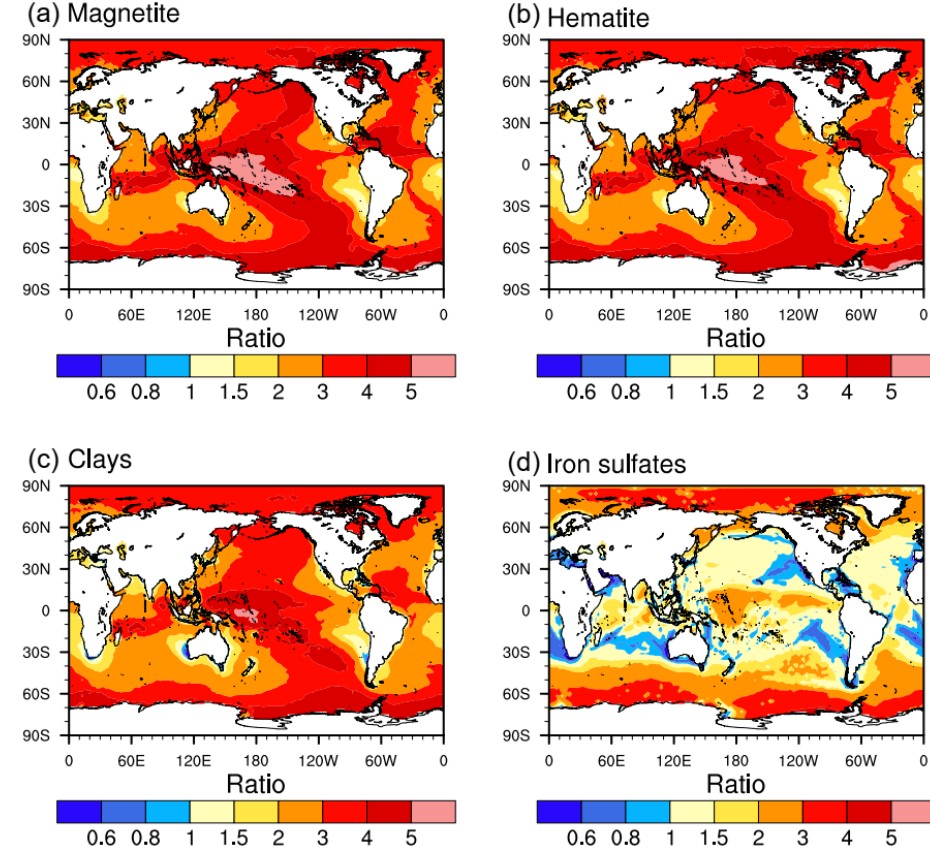


**Fig. 6**. Ratios of soluble iron deposition from four anthropogenic iron-containing minerals between the fine-sized and coarse-
sized cases.
In this study, the explicit treatment of anthropogenic iron mineralogy enables us to identify the iron mineral-dependent
variability. We find that for those coming primarily from fossil fuel combustion and iron smelting on land, namely magnetite,
hematite, and clays, the shift of iron emissions toward finer size bins promotes their long-range transport and enhances
corresponding soluble iron deposition to North Pacific, Equatorial Pacific, and Southern Ocean by more than a factor of 4 (Fig.
6a-c). However, the size distribution effects on iron sulfates are pronounced only in the polar areas, which are subject to plumes





of middle latitude shipping emissions. In line with previous study [*Rathod et al.*, 2022], the iron sulfates constitute an important
contribution to soluble iron deposition away from major continental sources, predominately associated with shipping emission
over local oceans rather than long-range transport from land (Fig. S5). Hence, the variability induced by iron size distributions
is less remarkable for iron sulfates than for other anthropogenic minerals. These results also suggest that the relative importance
of iron sulfates in total soluble iron deposition to remote oceans is altered by the size distributions of all other iron minerals
that originate from continental sources.

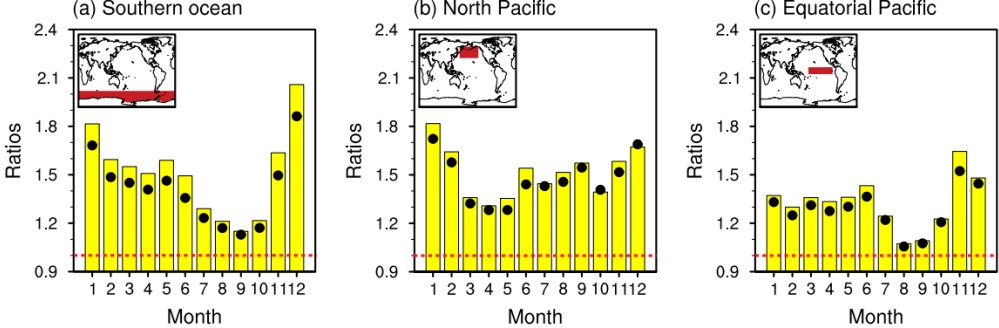


**Fig. 7**. Differences of monthly total soluble iron deposition between the fine-sized and coarse-sized groups over specific ocean
basins. Only anthropogenic iron emission sizes have been examined here. Histograms describe the ratio of monthly results in
the fine-sized case to that of the coarse-sized for (a) Southern Ocean, (b) North Pacific, and (c) Equatorial Pacific, respectively.
For comparison, black dots describe the ratio of the fine-sized results to that with global anthropogenic emission amount
scaling down by a factor of 2. The ocean basins of interest are indicated at the top-left corner of each panel. The red dashed
lines indicate the ratio of 1.0.

It is critical to examine monthly soluble iron availability altered by emission size distributions, because ocean primary
production can respond to iron inputs on the order of days to months [*Guieu et al.*, 2014]. From our results in Fig. 7, the
importance of anthropogenic emission size distributions in shifting soluble iron deposition varies by month over potentially
iron-limited ocean basins, i.e., HNLC regions, due to the episodic nature of natural iron sources (dust and wildfire) and their
deposition. For the Southern Ocean, the monthly ratios of the fine-sized case to the coarse-sized span from 1.1 in September
to 2.1 in December (Fig. 7a). The September peak of fire iron (shown in Fig. S6), possibly linked to low precipitation in
southern winter, masks the variability in anthropogenic iron contributions by emission size distributions. Conversely, the
largest difference in December is associated with the lowest contribution of natural sources (Fig. S6). By contrast, the monthly
differences are less fluctuated in North Pacific, ranging between 1.3 and 1.8 (Fig. 7b). Anthropogenic emission dominates
soluble iron throughout the year except in March-May, during which dust storms originating from East Asia frequently occur
and regulate soluble iron inputs to North Pacific. The Equatorial Pacific has the lowest ratio amongst the three regions, because
anthropogenic aerosol bearing plumes rarely arrive in this region and lots of rain out here can efficiently remove aerosols. For
the three ocean basins, such differences related to the emission size treatments are even larger than those by adjusting the iron
emission amount by a factor of 2 with the consideration of emission uncertainty (black dots in Fig. 7). We therefore suggest
that compared with iron emission fluxes, the representation of size distributions for anthropogenic iron is equally or even more
important to the estimation of total soluble iron deposition to remote oceans.



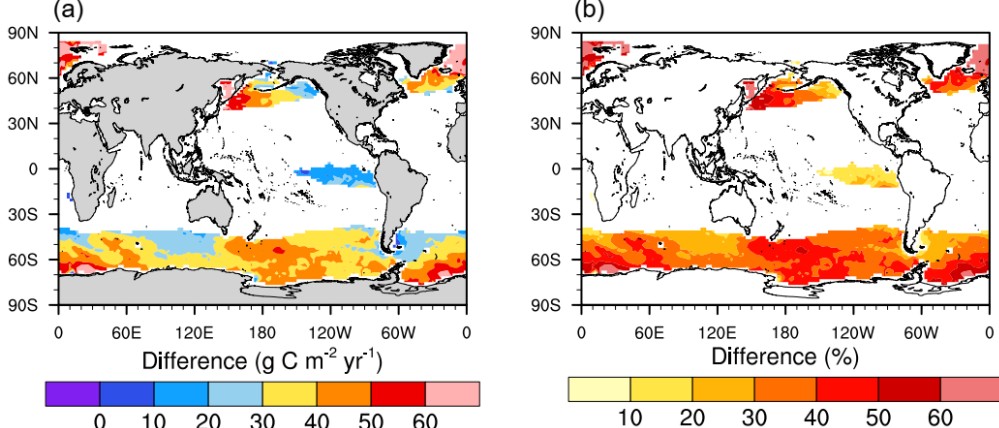


**Fig. 8.** Difference of marine net primary production sustained by atmospheric soluble iron between two iron size distribution
groups. Here, following *Rathod et al.* [2022], we focus only on the iron-limited ocean basins, which are defined using a cut-
off of nitrate concentrations at surface oceans.

We also provide an estimate of the changes in iron-sustained marine net primary production between the finer-sized and coarse-
sized groups (Fig. 8). In line with the distributions of soluble iron deposition, the effects of finer-sized iron distributions can
enhance primary production over remote oceans including the North Pacific Ocean and Southern Ocean as high as 50%.
Considering that anthropogenic iron aerosols may contribute to >10% of the total marine productivity in the North Pacific
Ocean [*Ito et al.*, 2020; *Rathod et al.*, 2022], the representation of their size distributions at emissions, mostly from East Asia,
is particularly important in the Earth system modeling. The evolution of atmospheric iron-aerosol size characteristics and their
emission fluxes can be critical players to ocean carbon sequestration from past to future. Hamilton et al. (2020) found that
historical air pollution controls has cut down anthropogenic emission amounts of particles in coarse sizes, in turn elevating the
mass fraction of finer-sized iron particles and thus the overall lifetime of atmospheric iron. Hence, the complex interactions of
iron and the Earth system is linked to human activity effects on soluble iron availability to ocean basins.

**4. Conclusion**
This study explores the extent to which iron size distribution at emission, specifically from anthropogenic sources, alters
estimates of soluble iron deposition to the open ocean. A global microphysical, size-resolved aerosol model is used to simulate
the iron cycle, involving emission, atmospheric processing, and deposition on a global scale. The model treats iron mineralogy,
size evolution, and chemical aging processes during atmospheric transport, which enables the investigation on the relationship
between iron size distributions and iron long-range transport and subsequent deposition. We test four representative size
distribution schemes for anthropogenic iron sources employed in previous studies.
We find that allocating a more balanced fraction of iron aerosol at emission into particle sizes less than 1 μm, results in a longer
atmospheric lifetime and mass burden of total iron aerosols by about a factor of 2 compared to a coarse-sized dominated case,
primarily associated with the decreased loss rates via dry and wet removal processes. The evaluation of anthropogenic iron
aerosols against the global-scale observation dataset reveals that despite the same emission fluxes considered in all cases, their
simulated magnetite aerosol concentrations differ by up to a factor of 10, while the higher fine-sized cases agree better with
the observations. It is therefore necessary to accurately represent iron size distributions in order to constrain iron emission
fluxes more realistically with aerosol simulations and observations [*Liu et al.*, 2022]. Our simulations show that the resulting
annual soluble iron deposition differs by up to a factor of 1.5 over remote oceans including the North Pacific Ocean and



Southern Ocean, because the fine-sized group allows more iron to be transported to a long distance with enhanced atmospheric
processing. More importantly, the monthly soluble iron deposition, which is relevant to ocean primary production responses
over days to months, would be enhanced by 110% and 80% in the fine-sized case over the Southern Ocean and North Pacific
Ocean, respectively. Such differences are similar to or even larger than those with the consideration of emission uncertainty,
suggesting the equally important role of iron size distribution treatment.
This study unravels the critical role of iron size distributions in shaping atmospheric soluble iron inputs to global oceans,
especially to the remote regions. However, the realistic understanding of iron emission size distributions is still inadequate
given limited observation data. Targeted in-site measurements on iron aerosol size along with its mass and solubility at source
areas are highly desirable. Furthermore, our finding may be extended to other key trace elements of importance to ocean
biogeochemistry, like copper, manganese, and phosphorus.

**Author contributions**: H.M. and M.L designed the research. M.L. performed model simulations, analyzed the data, and wrote
the original manuscript. M.L., H.M., D.S.H., S.D.R., K.D.L., and N.M.M. interpreted the results and discussed their
implications. All authors commented on and contributed to the manuscript.
**Competing interests**: The authors declare that they have no conflict of interest.
**Data availability**: The model data used to generate the figures can be available after the acceptance of the manuscript.
**Code availability**: The CESM source code is publicly available from NCAR at: https://www.cesm.ucar.edu/.
**Acknowledgement**
This study was supported by the Ministry of Education, Culture, Sports, Science, and Technology and the Japan Society for
the Promotion of Science (MEXT/JSPS) KAKENHI Grants (JP20H00196, JP22H03722, JP22F22092, JP22KF0165,
JP23H00515, JP23H00523, JP23K18519, JP23K24976, and JP24H02225); by the MEXT Arctic Challenge for Sustainability
II (ArCS II) Project (JPMXD1420318865); and by the Environment Research and Technology Development Fund 2–2003
(JPMEERF20202003) and 2–2301 (JPMEERF20232001) of the Environmental Restoration and Conservation Agency.
Mingxu Liu acknowledged the support of JSPS Postdoctoral Fellowships for Research in Japan (Standard). DSH was supported
by NASA (Proposal Number: 22-IDS22-0027). We acknowledged the NASA Radiation Sciences Program, the NASA Upper
Atmosphere Research Program, and the NOAA Atmospheric Composition and Climate Program for providing the aircraft
observational data.

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
