# Peer review of "Representation of iron aerosol size distributions of anthropogenic"

_EGUsphere, 2024_

## Author Comment (AC1)

**Response to Referee #1**

We appreciate Dr. Mónica Zamora Zapatas's helpful and constructive comments and accordingly improve our manuscript. The Referee's comments are marked in blue.

This study shows the effects of varying the iron emission size distribution on global-scale simulations to diagnose iron concentration in the atmosphere and ocean deposition. Four different size distributions are studied, properly justified, and their results are compared to aircraft observations, allowing them to conclude on the closest simulated distributions. The methods lack some details about the iron emission and deposition processes. Finally, some comments on the possible biases of using a global model could be useful, reflecting on the possibility of higher resolution models being helpful in further elucidating this phenomenon. Therefore, I recommend a minor revision. Some minor comments follow:

**Response**: Accepted. We add more description about the iron emission and deposition processes (Please see Line 104-107 and Line 92-93). The higher resolution models can be used at regions scales to explore the deposition and chemical aging processes over different polluted environments (Please see Line 345-346). Detailed revisions are shown in the following.

**Revisions**:

Line 92-93: "Both dry deposition (Zhang et al., 2001) and wet deposition (Liu et al., 2012) of aerosols are treated in our model. The improvement of aerosol in-cloud wet scavenging process was included to improve the modelling of aerosol long-range transport efficiency (Liu and Matsui, 2021)."

Line 104-107: "For anthropogenic iron, we consider five different minerals, namely magnetite, hematite, illite, kaolinite, and sulfate iron, following the global emission inventory by Rathod et al. (2020), which was developed by a bottom-up approach at 1° spatial resolution and 1-month temporal resolution. The reference year of the inventory was 2010. We account for a wide range of anthropogenic sources including iron smelting and fossil fuel combustion sources."

Line 345-346: "Higher resolution models with finer grids and detailed microphysics are useful to explore iron aerosol deposition and chemical aging processes at regional scales."

In the abstract, it would be great to highlight which findings are new.

**Response**: Accepted. This study, by performing a series of global aerosol simulations, demonstrates the importance of representing iron aerosol size distributions at emission in understanding global iron deposition to the ocean. This property has been overlooked in previous modelling analysis and should be well treated to narrow model uncertainties. Please see the revised statement at Line 21-26.

**Revisions:**

Line 21-26: "Shifting the iron aerosol size distribution toward finer particle sizes (<1 μm) enables longer atmospheric lifetime (a doubling), promoting atmospheric processing that enhances the soluble iron deposition to ocean basins by up to 50% on an annual basis. The monthly enhancements reach 110% and 80% over the Southern Ocean and North Pacific Ocean, respectively. Uniquely, our results highlight that compared with emission flux variability, iron emission size distribution plays an equally important role in regulating soluble iron deposition, especially to the remote oceans. Our new findings can help to interpret inter-model differences in iron deposition estimation and to better quantify the effects of atmospheric nutrients input on marine biogeochemistry, including but not limited to iron, phosphorus, and others."

L55 are all of these papers working with global models? Is there a variety of local to global approaches and if so, how does grid resolution and microphysics models differ?

**Response**: Accepted. All of those papers applied global models. Only global chemistry transport and general circulation models can be used to reflect the long-range transport effect on iron aerosol distribution. Therefore, we present them to explore why the size distribution of iron is a critical factor. Grid resolution and microphysics may be also important, but not the scope of this study, and we mention them in the discussion part (Line 345-346). Please see the revised sentence at Line 55-56 and Line 59-61.

**Revisions:**

Line 55-56: "To elucidate atmospheric flux of iron-containing aerosols to the ocean, global-scale aerosol models have been developed to include a range of iron emission sources that currently show a large intermodal difference in flux estimates (Myriokefalitakis et al., 2018)."

Line 59-61: "Among those global aerosol simulations, the size distribution of iron, which is an important consideration when determining aerosol lifetimes and thus its long-range transport potential, is key in shaping atmospheric iron distributions (Hamilton et al., 2020a; Myriokefalitakis et al., 2018)."

Line 345-346: "Higher resolution models with finer grids and detailed microphysics are useful to explore iron aerosol deposition and chemical aging processes at regional scales."

L118 Is it also possible that iron emissions have a different size distribution in other parts of the world? How could this simplification be considered for future work?

**Response**: Accepted. We tested four different size distributions for anthropogenic iron, all of which were applied on a global scale. This method is commonly used in previous modeling works. Applying region-specific size distributions may be more realistic but challenging due to limited knowledge about that. We add more discussions on this uncertainty. Please see Line 152-154 and Line 281-282.

**Revisions:**

Line 152-154: "To enable the intercomparison among these cases, we used the globalscale anthropogenic iron emission mass inventory from Rathod et al. (2020) but with different allocations between fine and coarse sizes in each case. The size distribution of anthropogenic iron emission in each case was treated uniformly on a global scale."

Line 281-282: "Because the size distributions of anthropogenic iron minerals may depend on different combustion processes, source- and region-specific size distribution representation is desirable in the future work."

**L95 I don't completely follow how the emissions are prescribed or calculated. Is it homogeneous over all surfaces in the world? Same with its absorption. Do they change over time?**

**Response**: We would like to state that in this paper, the emission means the amount of iron particles released into the atmosphere during the fossil fuel combustion and industrial activities. The iron emissions are provided by a grid-resolved emission inventory (Rathod et al., 2020). The emission fluxes are different in time and space around the world, with a temporal resolution of one month. The reference year of the inventory is 2010. Absorption is beyond the scope of this study. Please see Line 104-107.

**Revisions:**
Line 104-107: "For anthropogenic iron, we consider five different minerals, namely magnetite, hematite, illite, kaolinite, and sulfate iron, following the global emission inventory by Rathod et al. (2020), which was developed by a bottom-up approach at 1° spatial resolution and 1-month temporal resolution. The reference year of the inventory was 2010. We account for a wide range of anthropogenic sources including iron smelting and fossil fuel combustion sources."

**Fig. 4,5,6: Are these plots derived from yearly averaged values?**

**Response**: Accepted. Yes, we describe that these plots are made using either yearly averaged or yearly accumulated model results. Please see the revised figure captions for Figs. 4, 5, 6.

Reference:

Rathod, S. D., Hamilton, D. S., Mahowald, N. M., Klimont, Z., Corbett, J. J., and Bond, T. C.: A Mineralogy-Based Anthropogenic Combustion-Iron Emission Inventory, J. Geophys. Res.-Atmos, 125, e2019JD032114, 10.1029/2019jd032114, 2020.

---

## Author Comment (AC2)

**Response to Referee #2**

This manuscript reports on a numerical modelling study that examines the impact of the size distribution of iron-bearing anthropogenic emissions on the transport and deposition of soluble iron to the oceans. The paper is extremely well written and the work is presented clearly and concisely. I have no hesitation in recommending that it is suitable for publication in ACP with only minor revisions.
**Response**: We appreciate the reviewer's positive comments and make minor revisions as follows.

Specific Comments

Line 1. The work reported focusses exclusively on the size distribution at emission of anthropogenic iron. The title of the paper does not reflect this. I suggest that "Representation of iron aerosol size distributions of anthropogenic emissions is critical in evaluating atmospheric soluble iron input to the ocean" would give a better representation of the subject of the paper.
**Response**: Accepted. We revise the title as: Representation of iron aerosol size distributions of anthropogenic emissions is critical in evaluating atmospheric soluble iron input to the ocean.

L 84. Given the dominance of dust as a source of Fe, it might be helpful for the reader to be given a little more detail on the representation (and validation) of the size distribution of dust Fe in the model. Similarly, the model described identifies various Fe-bearing minerals in anthropogenic emissions (L 100 – 101). Presumably at least some of these minerals also occur in dust. How does the model treat these minerals in dust, and how are the anthropogenic and dust fractions handled in the comparison to the aircraft observations (e.g. Fig. 3)?
**Response**: Accepted. Following these points, we revise the manuscript below.
1) We add more details on the treatment of dust Fe size distribution in our model. As a part of dust aerosols, dust Fe features the same size distribution as dust, derived by Kok (2011). Please see Line 108-110 in the revised manuscript.
2) As commented by the reviewer, this study focuses on anthropogenic iron emission, and natural dust mineralogy was not considered in our simulations. Though some minerals, e.g., magnetite, can come from both anthropogenic and dust emissions, the dust fractions have been excluded from the aircraft observations (by Lamb et al., 2021). So that we can use them directly for comparison with our anthropogenic iron simulations results. We add such sentences at Line 113-116.

**Revisions**:
Line 108-110: "Dust iron emission was calculated by assuming a constant iron content of 3.5% in dust aerosol emission (Shi et al., 2012). The model estimated total dust emission fluxes using the scheme of Zender et al. (2003), with modifications by Albani et al. (2014) and the size distribution from Kok (2011)."

Line 113-116: "We validated our modeled anthropogenic iron oxide concentrations against a global-scale aircraft measurement in the troposphere consisting of eight campaigns for the periods of 2009–2011 and 2016–2018 (Lamb et al., 2021). These observations provide mass concentrations of anthropogenic iron oxide, i.e., magnetite, with volume equivalent diameters between 180-1290 nm."

L 96-97. Please comment on the implications of the assumption of internal mixing of Fe with all other aerosol components for your results.

**Response**: Accepted. This assumption is reasonable for anthropogenic iron aerosols, which are often mixed together with other aerosol compounds in the polluted environments, e.g., East Asia (Li et al., 2017). The internal mixing enables rapid growth of small aerosols into larger ones via condensation and coagulation and makes them deposited more efficiently than aerosols in external mixing state. Please see our added description at Line 100-102.

**Revisions**:
Line 100-102: "The internal mixing assumption is reasonable for anthropogenic iron aerosols, which are often mixed together with other aerosol compounds in the polluted environments, e.g., East Asia, and enable the growth of iron aerosols via condensation and coagulation."

L 189 – 190. Please explain this statement further.

**Response**: Accepted. We would like to state that the size range of aerosols for comparison should be consistent between observation and simulation. The measurement data is available for sub-micron mode of aerosols (i.e., 180-1290 nm in diameter), and correspondingly the model results with the similar size range were extracted. Please see the revised description at Line 113-116 and Line 197-198.

**Revisions**:
Line 113-116: "We validated our modeled anthropogenic iron oxide concentrations against a global-scale aircraft measurement in the troposphere consisting of eight campaigns for the periods of 2009–2011 and 2016–2018 (Lamb et al., 2021). These observations provide mass concentrations of anthropogenic iron oxide, i.e., magnetite, with volume equivalent diameters between 180-1290 nm."
Line 197-198: "We extracted the modeled mass concentrations of iron aerosols with the size range similar to that of the measurements."

L 223 – 224. Perhaps some clarification is needed here? The differences referred to do appear to be more pronounced over much of the global ocean, but not over the Southern Ocean.

**Response**: Accepted. We reword this sentence and clarify that the differences appear to be more pronounced over much of the global ocean, except the Southern Ocean. Please see Line 230-231.

**Revisions**:
Line 230-231: "By contrast, the differences for soluble iron are more pronounced than for total iron over much of the global Ocean (Fig 4c), because of the higher solubility of anthropogenic iron than dust iron."

L 225 – 227. Is the solubility enhancement referred to here relevant to dust Fe, anthropogenic Fe, or Fe in general?
**Response**: Accepted. This refers to anthropogenic Fe. Over East Asia, anthropogenic iron emissions are pronounced and subject to efficient aging processes during transport. Please see Line 231-233.

**Revisions:**
Line 231-233: "Over East Asia and its outflow areas, the rapid aging process in the polluted environments are capable of enhancing iron solubility, particularly those of anthropogenic origin"

Technical Corrections

L 102. Delete "taken into account".
**Response**: Deleted. Please see Line 106-107.

L 117. Also add " of anthropogenic emissions" to this section heading?
**Response**: Corrected. Please see Line 124.

L 139. "A similar…"
**Response**: Corrected. Please see Line 146. -thanks for correcting typos.

L 190. "… used as an …"
**Response**: Corrected. Please see Line 199.

L 250. "iron source", rather than "iron emission"?
**Response**: Corrected. We reword it as "iron source". Please see Line 257.

Fig. 8. Please add a description of the two panels. (b) shows a percentage difference. What is this difference relative to?
**Response**: Accepted. We add a description of the two panels as: The panels display (a) absolute differences and (b) percentage differences in net primary production of the fine-sized group relative to the coarse-sized group.
Please see the revised caption below Fig. 8.

References:
Kok, J. F.: A scaling theory for the size distribution of emitted dust aerosols suggests climate models underestimate the size of the global dust cycle, Proceedings of the

National Academy of Sciences of the United States of America, 108, 1016, 10.1073/pnas.1014798108, 2011.

Li, W., Xu, L., Liu, X., Zhang, J., Lin, Y., Yao, X., Gao, H., Zhang, D., Chen, J., Wang, W., Harrison, R. M., Zhang, X., Shao, L., Fu, P., Nenes, A., and Shi, Z.: Air pollution–aerosol interactions produce more bioavailable iron for ocean ecosystems, Sci. Adv., 3, e1601749, 10.1126/sciadv.1601749, 2017.

Lamb, K. D., Matsui, H., Katich, J. M., Perring, A. E., Spackman, J. R., Weinzierl, B., Dollner, M., and Schwarz, J. P.: Global-scale constraints on light-absorbing anthropogenic iron oxide aerosols, npj Clim. Atmos. Sci., 4, 10.1038/s41612-021-00171-0, 2021.